# Preoperative Anxiety in Patients with Pancreatic Cancer: What Contributes to Anxiety Levels in Patients Waiting for Surgical Intervention

**DOI:** 10.3390/healthcare11142039

**Published:** 2023-07-17

**Authors:** Veronica Marinelli, Maria Angela Mazzi, Michela Rimondini, Olivia Purnima Danzi, Deborah Bonamini, Claudio Bassi, Roberto Salvia, Lidia Del Piccolo

**Affiliations:** 1Department of Surgery, Dentistry, Pediatrics and Gynecology, University of Verona, 37134 Verona, Italy; 2Department of Neurosciences, Biomedicine and Movement Sciences, University of Verona, 37134 Verona, Italy

**Keywords:** preoperative anxiety, pancreatic surgery, anxiety scales, oncological surgery, pancreatic cancer, psychological support, psycho-oncology

## Abstract

Pancreatic cancer is one of the most lethal malignancies. Currently, the only treatment is surgical resection, which contributes to significant preoperative anxiety, reducing quality of life and worsening surgical outcomes. To date, no standard preventive or therapeutic methods have been established for preoperative anxiety in pancreatic patients. This observational study aims to identify which patients’ socio-demographic, clinical and psychological characteristics contribute more to preoperative anxiety and to identify which are their preoperative concerns. Preoperative anxiety was assessed the day before surgery in 104 selected cancer patients undergoing similar pancreatic major surgery, by administering the STAI-S (State-Trait Anxiety Inventory Form) and the APAIS (Amsterdam Preoperative Anxiety and Information Scale). Our data suggest that patients with high STAI-S showed higher levels of APAIS and that major concerns were related to surgical aspects. Among psychological characteristics, depressive symptoms and trait anxiety appeared as risk factors for the development of preoperative anxiety. Findings support the utility of planning a specific psychological screening to identify patients who need more help, with the aim of offering support and preventing the development of state anxiety and surgery worries in the preoperative phase. This highlights also the importance of good communication by the surgeon on specific aspects related to the operation.

## 1. Introduction

Pancreatic cancer (PC) is a highly lethal malignancy, with an overall 10% five-year survival rate [1]. The incidence of PC is increasing worldwide, whereas therapeutic options remain limited, mainly due to the fact that the diagnosis is made at an advanced stage of the disease. Accordingly, PC is projected to become the second leading cause of cancer-related deaths in 2030 [2]. Surgical resection with adjuvant systemic chemotherapy currently provides the only chance of long-term survival [3].

Surgical intervention is a traumatic treatment method that causes major life changes and leads to anxiety in patients [4]. Anxiety experienced in the preoperative period is defined as the feeling of uncertainty, restlessness and fear associated with hospitalization, surgery and anesthesia [5]. Most patients perceive the day of surgery as the biggest and most threatening day in their lives, and 11% to 93% experience preoperative anxiety because of their feeling of uncertainty, fear of disability and death [6,7].

In a meta-analysis, Abate et al. [5] evaluated 14,652 surgical patients in 17 countries and found that the universal preoperative anxiety prevalence was 48% (95% CI: 39–57%). There are also copious studies that indicate that a high level of preoperative anxiety negatively affects the intervention itself and the subsequent recovery. High anxiety levels lasting for a prolonged time have been shown to affect neuroendocrine functions, increasing the need for anesthesia and analgesics [8,9,10], intensifying post-operative pain severity [9,10], lengthening recovery time from anesthesia [11], increasing post-operative mortality [8,9] and contributing to postoperative delirium [12].

Anticipation of postoperative pain, separation from the family, incapacitation, loss of independence and fear of surgery and death are factors that trigger symptoms of perioperative anxiety [13]. In addition, the extent of this distress may be influenced by previous psychiatric diseases, such as depression, anxiety, and minor psychiatric disorders [14,15]. Recent studies by Barnes et al. [16] and Del Piccolo et al. [17] reported that depression is raised in people with pancreatic cancer compared with the general population, confirming that exploring and taking care of pancreatic patients’ emotional distress must be considered a top priority in order to protect their quality of life (QoL). Thus, appropriate and careful screening of the psychological conditions of patients undergoing major pancreatic surgery may contribute to identifying patients who need more help, with the aim of offering support and preventing the development of emotional distress, state anxiety and surgery worries. Nevertheless, despite pancreatic patients showing a significant psychological burden related to their condition, no specific description of the factors related to emotional distress and, more specifically, to preoperative anxiety levels has been reported to date in the literature. Finally, recent studies show that a greater amount of information received prior to surgery does not always contribute to reduced anxiety levels [18,19], whereas shared and positive information, delivered in an empathic environment, can reduce preoperative anxiety and increase surgical recovery [20].

Therefore, the aim of the present paper is twofold:To identify patients’ socio-demographic, clinical and psychological characteristics that most contribute to preoperative anxiety. According to Spielberg, anxiety can be distinguished into trait anxiety, as a stable feature of personality, and state anxiety, as the degree of anxiety at a particular time [21]. The latter is the type of anxiety considered in our study, and defined as preoperative anxiety preceding pancreatic surgery.To explore the link between patient’s preoperative anxiety and the type of preoperative concerns related to surgery. Identifying which are the key concerns to address can help surgeons to improve doctor-patient communication.

Figure 1 reports graphically the logic underlying research questions.

## 2. Materials and Methods

### 2.1. Study Design and Participants

This observational study is part of a larger research project, carried out at the inpatient clinic of the Pancreas Institute of the University Hospital of Verona (AOUI), Italy, which is the first Italian center entirely dedicated to diagnosis, treatment, and research on pancreatic diseases. Here we explored a homogeneous group of 104 patients, all undergoing major surgery for pancreas cancer, between June 2017–May 2018, who have been picked out from the PREPARE RCT sample of 114 patients [22] (clinicaltrial.gov identifier: NCT03408002). As indicated in Appendix A, 10 patients were excluded because of palliative surgery. The day before surgery, all patients underwent a visit to the surgeon who delivered information and to the clinical psychologist who provided support and indications on how to manage anxiety and main concerns. Written informed consent was obtained from all patients who agreed to participate in the study. Study approval was obtained from the Verona Research Ethics Committee (Prog. 1288CESC).

### 2.2. Measures

A sociodemographic schedule was applied, collecting gender, age, scholarship, family status and employment, geographical area of origin, clinical condition, disease status and type of intervention to undergo.

The Italian adaptation [23] of the common and widespread used State-Trait Anxiety Inventory Form Y (STAI-Y) [21], which evaluates both levels of dispositional/trait anxiety (how one feels usually, STAI-T) and state anxiety linked to a specific event/time window (STAI-S), was administered in two distinct time sections: the STAI-T form was administered during day-hospital preoperative counseling, usually 1 month before surgery, and STAI-S at the time of hospitalization, usually the day before the intervention. In both forms, it is a self-assessment questionnaire composed of 20 items on a Likert scale ranging from 1 (not at all) to 4 (very much), so that the total score is the sum of 20 items, ranging from 20 to 80; the higher the values, the greater is the level of anxiety. Appendix A presents the frequency distribution of each item of the STAI-S, which is used in this study to measure the prevalence of preoperative anxiety. In the present study sample, internal consistencies (Cronbach α) were high either for STAI-T (α = 0.86) or STAI-S (α = 0.93). There are no published specific normative STAI-S values for hospital cancer patients, therefore, for prevalence calculation, in our study, we followed Bunevicius et al. [24] indications (in their study on cardiac patients they found that the STAI-S had a sensitivity of 89% and a specificity of 56% when tested against other known measures for clinical anxiety) and applied a cut-off score of STAI-S ≥ 45 to represent clinical anxiety. Moreover, due to the high mean age of our sample, we also adopted Ilardi et al. [25] cut-off score > 61 on the elderly population as an indication for a score outside the tolerance limit (90th percentile—severe anxiety).

Patient Health Questionnaire (PHQ-9, [26]) is a self-report questionnaire specifically used in primary care for screening, diagnosis, monitoring and measurement of depression severity. The PHQ-9 consists of 9 items corresponding to the symptoms of major depression, according to the Diagnostic and Statistical Manual of Psychiatric Disorders (DSM-V, [27]). Its sensitivity and specificity are recognized as optimal to highlight depression of clinical relevance. The score ranges between 0 and 27 points, with cut-offs set to indicate the severity of depression: (1) 5 to 9, minimum depressive symptoms/subthreshold depression; (2) 10 to 14, minor depression/minor major depression; (3) 15 to 19, moderate major depression; and (4) over 20, severe depression.

The Amsterdam Preoperative Anxiety and Information Scale (APAIS, [28]) is a quick and easy questionnaire, composed of six items taking notice of the patient’s concerns about the surgical intervention on the Likert scale. The Italian version has been translated and validated by Buonanno et al. [29]. Regarding the agreement scale extent, the original version has five response categories, ranging from “not at all” (1) to “extremely” (5), where 3 is the neutral score, the total score ranges between 6–30 and the threshold cut-off is 14. To force respondents to express their perspective and to avoid a neutral position [30], we preferred to adopt an even number of response categories (6 values), that changed the total score range to 6–36, with a rescaled threshold cut-off of 17.4. To check the internal consistency of the scales we performed Cronbach’s Alpha (see Appendix A). Following the approach proposed by Aust et al. [6], attention is paid to the magnitude of each component of patient anxiety related to anesthesia (APAIS-A), surgery procedure (APAIS-S) and the information needs (APAIS-I). The subscales are obtained by summing up the response scores as follows: items 1 and 2 are merged into APAIS-A, items 4 and 5 into APAIS-S, and the remaining items 3 and 6 into APAIS-I (see Appendix A). The Pearson correlation values between the three APAIS subscales were low (from 0.22 to 0.36), indicating that they investigate different topics (see Appendix A).

### 2.3. Statistical Analysis

Preliminary analyses were performed in order to describe frequency distribution (i.e., mean standard deviation, skewness and kurtosis) and to check outliers and missing data. Bivariate exploration (i.e., ANOVA approach) was then performed on the two primary outcomes (i.e., APAIS and STAI-S) in order to check differences among sub-groups. The results were also checked using a non-parametric approach (median test for equal distribution) to verify the violation of the normality assumptions of the target variables.

Since APAIS and STAI-S were gathered on the same day, Jonckheere–Terpstra test for trend [31] and dominance approach [32] to regression were adopted to check the monotonic trends of each APAIS component in relation to the different scores of STAI-S.

To verify the second research question of our study (to explore the link between patient’s preoperative anxiety and the type of preoperative concerns related to surgery indicated by the mediator role of APAIS), a multivariate analysis was then performed by using path analysis, to estimate two different models explaining the relationships between STAI-T, PHQ-9, APAIS components and STAI-S scores. The path analysis approach can be considered as an extension of multiple regression since it allows us to represent the relationships in terms of process, and to assess direct and indirect effects [32,33]. Two models were then elaborated: 1. a basic model (H_0_ hypothesis) which fitted just the direct effect of preoperative concerns (APAIS) on preoperative state anxiety (STAI-S), considering STAI-T and PHQ, as covariates; 2. a second model (H_1_ hypothesis) which assumes that preoperative concerns (APAIS) mediate the effects of patient psychological characteristics (STAI-T and PHQ) on preoperative state anxiety (STAI-S) (see Appendix A).

Path analysis, based on the structural equation framework, was run with SEM command, using Satorra-Bentler adjustments. This accounts for violations of assumptions, such as heteroskedasticity and non-normal distribution of endogenous variables. Post-hoc explorations followed, by using the MEDSEM command [34], to explore the presence of significant partial or full mediation effects, based on the bootstrap approach. This approach allowed us to estimate a bias-corrected interval confidence [35]. Given the small dimension of our sample, we also performed a power calculation using Power4SEM, a free R package software [36] (see Appendix A).

Stata 17 software was used for all analyses.

## 3. Results

The 104 cancer patients included in this study underwent the following types of major pancreatic surgery: 54% DCP–whipped, 25% partial spleno-pancreatic resection, 13% exploratory laparotomy and 11% total spleno-pancreatic resection. Their median age was 65 years (range: 26–79; mean = 63, sd = 11.5), and females were 54%. More details of sociodemographic and clinical frequency distributions are reported in Table 1 (column 1).

At the time of admission to the Unit of General and Pancreatic Surgery, 41 (39%; 95% CI = 30.0–49.5) patients had clinically relevant levels of anxiety (STAI-S cut-off ≥ 45 [17]), among them 15 (14%) had severe anxiety, according to Ilardi’s criterion [25]. The median score for state-anxiety STAI-S was 42 (range 33.5–49.5). Appendix A reports the frequency distribution of the responses to each STAI-S item.

Regarding preoperative concerns, as investigated by APAIS, 31 patients (30%; 95% CI = 21.2–40.0) showed a score exceeding 17.4 fixed thresholds. More specifically, the fear of surgery procedure seemed to play a prevailing role (47% of the APAIS total score—as can be seen in Appendix A, 53% and 42% of responders selected high scores > 4 for the two items—4 and 5—composing the surgery procedure subscale), followed by anesthesia fears (30%) and the need of more information (23%) as indicated by the means reported in the first row of Table 2.

Through dominance analysis, the regression model estimated that APAIS subscales explained 40% of the variability of STAI-S. Specifically, the STAI-S was more linked to surgery worries (R^2^ = 32%, corresponding to 81% of standardized dominance), followed by worries of anesthesia (R^2^ = 5%, corresponding to 12%) and information needs (R^2^ = 3%; 7%). Figure 2 shows the strength of the connection between APAIS components and STAI-S scores: “surgery” component has a steep slope (Jonckheere–Terpstra test for trend z = 6.23; *p* < 0.01), the “anesthesia” a moderate one (z = 3.05; *p* < 0.01) and the “more-information” component appears almost flat (z = 1.40; *p* = 0.16).

Comparisons according to demographic and clinical characteristics of participants on STAI-S scores and APAIS components showed statistically irrelevant differences, whereas differences emerged for STAI-T and PHQ-9 (see Table 1 and Table 2). Patients with higher trait anxiety (STAI-T) and more symptoms of depression (PHQ-9) showed greater levels of preoperative anxiety (Pearson rho of STAI-S was 0.55 with STAI-T and 0.41 with PHQ-9) and more worries linked to surgical intervention (Pearson rho of APAIS-S was 0.38 with STAI-T and 0.26 with PHQ-9 scores; APAIS-A were: 0.30 and 0.31; APAIS-I were: 0.23 and 0.33 respectively).

As indicated in Table 1 and Table 2, gender, age and education showed no significant impact on STAI-S and APAIS scores. A difference related to gender was only observed on trait anxiety (STAI-T), with higher mean scores in the female group (mean value 33.2, sd 9.3, vs. male: 29.1, sd. 7.7, t = 2.45 *p* = 0.02), therefore we preferred not to account for gender confounding effect in path analysis models, leaving gender differences subsumed under trait anxiety (STAI-T) scores.

Moving to a multivariate approach, we compared the two models (see Appendix A) in order to verify whether APAIS mediates the effect of psychological characteristics on Preoperative State Anxiety (STAI-S). Following a parsimonious criterion, we reported in Table 3 significant pathways identified by SEM analysis: only one of the 3 APAIS subscales (i.e., APAIS surgery) plays a mediator role on STAI-S in our sample. The H_1_ model seems to fit well and it is consistent with data (Chi_2_(8) = 17.19, *p* = 0.03; RMSEA_SB_ = 0.08; CFI_SB_ = 0.95; TLI_SB_ = 0.91; AIC = 3516.53; BIC = 3566.41; CD = 0.44).

The post-hoc analyses, expressed as conditional processes, confirmed that trait anxiety (STAI-T) can have an impact on preoperative concerns which in turn influence preoperative anxiety (STAI-S). More specifically, trait anxiety seems to play both a direct (Unstandardized Direct Effect UDE = 0.45) and indirect effect (Unstandardized Indirect Effect UIE = 0.25; *z*-test = 3.8 *p* < 0.01) on preoperative state anxiety, via worry for the surgical procedure, which plays a role of “mediator” (RIT = 35%). Depression, instead, maintains a moderate direct effect on preoperative anxiety, showing also a high comorbidity with STAI-T (with an estimated correlation of 0.40).

Figure 3 shows the effect of surgery related concerns (APAIS-S) on preoperative anxiety by distinguishing patients who have high (n = 21) and low (n = 82) affective disorders on the basis of both STAI-T and PHQ-9 evaluations (STAI-T ≥ 40 or PHQ-9 ≥ 10) as assessed during day-hospital counseling.

## 4. Discussion

Anxiety is a universal normal response to life-threatening procedures such as surgery and anesthesia [6,10]. Nevertheless, there are patients who are more affected than others by pre-operative anxiety. Therefore, determining significant predictors of preoperative anxiety can help healthcare providers to early identify those who need more help and support by implementing interventions that can prevent conditions of serious discomfort. For this reason, we conducted an observational study that aimed both to determine the preoperative anxiety level in pancreatic surgical participants the day before surgery (STAI-S score and proportion of patients with clinically significant preoperative anxiety), and to identify which socio-demographic, clinical and psychological characteristics, together with the analysis of the type of concerns related to surgery (APAIS), contributed most to preoperative anxiety. This second aim implied the use of longitudinal data (collected during the day-hospital stay, about one month before hospitalization for surgery procedure), which were considered as psycho-social predictors of preoperative anxiety, and cross-sectional cognitive-affective information (APAIS) that functioned as a possible mediator of a path analysis predictive model. The introduction of this last variable is based on cognitive behavioral models, which underly how the identification of key concerns (cognitions, thoughts and worries) may contribute to better recognizing the mental components related to preoperative state anxiety. Understanding these cognitive components can help health providers to become more aware of which topics to focus on when talking with patients, contributing to improving their effectiveness in managing communication and providing information. Given the importance of these cognitive aspects, to better accentuate the polarization of patient responses on APAIS items, we chose to force respondents to express their perspectives and to avoid a neutral position [30]. In this way, specific attitudes towards the three main dimensions describing presurgical worries in the APAIS scale could emerge more clearly. This choice has not led to significant consequences in terms of internal consistency (as evidenced by Cronbach’s Alpha indexes which were generally acceptable or good). Total Cronbach’s Alpha in our study was fairly good (Cronbach’s Alpha 0.73) even if slightly lower than that reported by the author of the Italian validation [29] (Cronbach’s Alpha 0.84) on a sample of 110 patients undergoing elective surgery. The second aim was accomplished by testing two possible path analysis models: 1. a basic model (H_0_ hypothesis) which fitted just the direct effect of preoperative concerns (APAIS) on state anxiety, considering STAI-T and PHQ, as covariates; 2. a second model (H_1_ hypothesis) which assumed that preoperative concerns (APAIS) mediated the effects of patient psychological characteristics (STAI-T and PHQ) on preoperative state anxiety. Since the sample was numerically limited (104 subjects), all statistical assumptions that would allow for obtaining reliable results were carefully verified.

Concerning the main results, in our study we evaluated trait (STAI-T) and state anxiety (STAI-S) at two different moments: STAI-T was gathered during day-hospital evaluation, about one month before hospitalization, and STAI-S was collected the day before surgery. Interestingly, anxiety trait levels were substantially low during day-hospital preoperative counseling (87 patients (83.7%) showed trait anxiety levels under the clinical threshold of STAI-T ≥ 40) but, in reverse, on the day before surgery 39% of the participants had raised levels of anxiety according to STAI-S. Literature reports that the prevalence of anxiety prior to any surgery ranges from 11 to 80% among adults [37,38]. As indicated in the introduction, Abate et al. [5] found that the universal preoperative anxiety prevalence in surgical patients in 17 countries was 48%, showing that our data are in line with the main literature. However, it should be noted that, even if our sample showed to cope well with illness during day-hospital assessment, the day before surgery the number of patients with high levels of arousal and worry increased significantly, given the complexity of the intervention. Interestingly, we observed also that in our sample apparently less serious patients (with a diagnosis at the entrance of uncertain biological behavior of the pancreatic disease) showed greater apprehension during the investigation of trait anxiety. Medical uncertainty increases anxiety, and it seems that the idea of possibly undergoing major surgery increases distress, especially in the less severe patients. Those who have less chance of survival because of the presence of malignancies perceive surgery as the only healing opportunity in which hope strongly compensates for uncertainty. This may also contribute to explaining why our percentages of preoperative anxiety are in line with mean values reported in the literature and not higher, as could be expected by the complexity of the intervention that these patients have to undergo.

Regarding gender, age and education, no significant differences were found in our primary outcomes related to preoperative anxiety (STAI-S and APAIS scores). A difference related to gender was observed only on trait anxiety (STAI-T), with higher mean scores in the female group. Nevertheless, gender did not affect directly STAI-S and APAIS scores, therefore we preferred not to account for gender confounding effect in path analysis models, given that gender differences could be subsumed under trait anxiety (STAI-T) scores. This has also practical implications: thinking about a screening activity during a day-hospital stay, it is preferable to stick to the use of specific screening tests rather than to predict outcomes on the basis of personal characteristics such as gender, with the risk of suggesting evaluations based on gender stereotypes.

Regarding other variables collected in day-hospital, no group differences were observed for depressive symptoms as measured by the PHQ-9 scale. As could be expected, we found high comorbidity between anxiety and depressive symptoms, showing that these clinical conditions frequently overlap, at least in this population [39] and both need to be considered in a screening evaluation during the day-hospital stay. A recent study by our group showed that patients, candidates for pancreatic resection [17], often present depressive symptoms related to demoralization and physical exhaustion, self-blame, denial and disengagement in coping. These are drivers of emotional distress and low self-efficacy, which in turn may contribute to triggering anxiety symptoms.

Together with STAI-S, the day before surgery we collected also the APAIS. As could be expected and similar to previous studies [40,41], we found a highly positive correlation between the two scales, which, respectively, represent the affective (STAI brings out the emotional and physiological correlates of anxiety) and the cognitive (APAIS brings out the type of concerns related to surgery) sides of preoperative anxiety. According to cognitive behavioral models, beliefs and thoughts impact and shape the quality of affective states (anxiety), which can then be modulated by restructuring the content of personal cognitions. In a recent qualitative study, King et al. [42] investigated preoperative anxiety components related to surgery, showing that patients experienced concerns such as uncertainty about the surgical process, fears related to possible complications such as postoperative infection, death during surgery, loss of privacy control, the possibility of being diagnosed with malignant cancer, experiencing postoperative pain, not being able to take care of family members or to return to daily life. Our findings confirm the observations reported by King [42], evidencing the role of concerns directly related to surgery procedure (APAIS-S). The findings showed that the fear of surgery detected by the APAIS scale (mainly the APAIS-S surgery component) significantly affected state anxiety (STAI-S) by playing an intermediating (moderating) role. This suggests that appropriate training of surgeons in communications skills and correct identification of the main topics to discuss during preoperative phases may contribute to possibly lower higher levels of state anxiety. This seems to be further confirmed by the results emerging from multivariate regression and path analysis, which showed that trait anxiety (STAI-T) and depressive symptoms (PHQ-9) influenced preoperative concerns (APAIS) and accordingly, preoperative state anxiety (STAI-S). It has been suggested that reducing anxiety during a consultation may lead to better retention of information, a stronger physician-patient relationship, and, ultimately, enhanced well-being [43]. Physicians need to be able to use effective communication skills: effective use of assessment, information and supportive skills can contribute to reducing patient anxiety [44]. Similarly, other professionals may contribute to lower preoperative anxiety, as suggested by a recent study published by our group [22]. This more recent work showed that even a brief psychological intervention in a dedicated session can increase a patient’s self-efficacy and awareness in addressing anxiety symptoms before major pancreatic surgery. Other studies reported that informing the patient about surgery and anesthesia is the first stage of the patient’s psychological preparation for surgery, and it is an important factor to reduce preoperative anxiety [11,41]. Therefore, early identification of patients’ concerns is crucial in reducing preoperative anxiety and in accelerating the postoperative recovery process. To help this process, an initial screening, when administering the informed consent during a day hospital stay, may be useful to identify patients who need more help, with the aim of offering support and preventing the development of state anxiety and surgery worries in the preoperative phase. Also, the introduction of a protocol such as the six-stage Spike-A proposed by Baile and Buchman [45], or the revised version proposed by Meitar and Karnieli-Miller [46] can contribute to developing specific procedures during the consent process for surgery to reduce preoperative anxiety, increase the quality of information and strengthen patient satisfaction.

This study had some limitations. It was a single-center study, which enabled us to evaluate preoperative anxiety in patients who were candidates for pancreatic surgery at a teaching hospital, but data cannot be generalized to all general hospitals. Patients undergoing emergency surgery were not included. Only elective surgical participants and only those who could be evaluated by the psychologist the day before surgery were considered. But we have also to remember that upfront surgical resection of pancreatic cancer is feasible in less than 20% of patients at diagnosis [47], therefore high numbers in this kind of sample are difficult to reach. For this reason, future studies need to include different hospital settings.

Another limitation relates to the variables included as predictors. Other psychological characteristics, such as personality traits were not assessed in this study. Beneath they are related to mental health [48], we have to consider that the study was based on screening activities conducted during a day-hospital stay, therefore the analysis of personality characteristics was beyond the scope of routine hospital activity. Next studies may include this variable as a further element of interest, when analyzing the relationship between emotional state, mental health and worries related to surgery. Finally, to better support our inferences on final models, it could be of interest to study a more defined temporal relationship between cognitive concerns and preoperative anxiety, to better understand the causal relationship between the two.

## 5. Conclusions

Anxiety in the perioperative period has a significant impact on both the flow of surgery and the post-operative recovery process. Our data show that patients with high state anxiety (STAI-S) showed higher levels of perioperative anxiety (APAIS) and that the major concerns were related to surgical aspects. Among psychological characteristics, depressive symptoms (PHQ-9) and trait anxiety (STAI-T) appeared both as risk factors for the development of preoperative anxiety, and were also very often correlated during day-hospital screening.

Findings from this study indicate that it is possible to identify those patients at higher risk of developing preoperative anxiety, suggesting that planning a specific psychological screening may help to assess those who need more help to prevent distress in the preoperative phase. Furthermore, based on the mediating role of surgery concerns on state anxiety we think that improving communication on surgery aspects can contribute to supporting patients and lower their concerns related to preoperative anxiety. Thus, also surgeons can contribute greatly to patient’s wellbeing by following appropriate communication protocols. Therefore, we suggest routinely assessing preoperative anxiety levels of patients using surgery-specific measurement tools to determine the worries associated with surgery and to reduce anxiety levels by adopting patient-centered medical care communication and practices.

## Figures and Tables

**Figure 1 healthcare-11-02039-f001:**
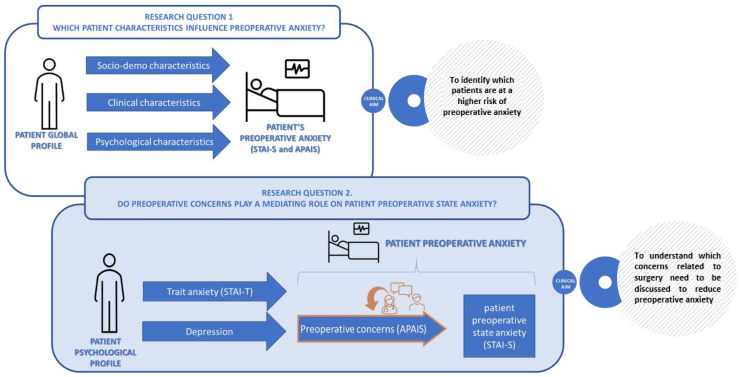
Graphical representation of research questions.

**Figure 2 healthcare-11-02039-f002:**
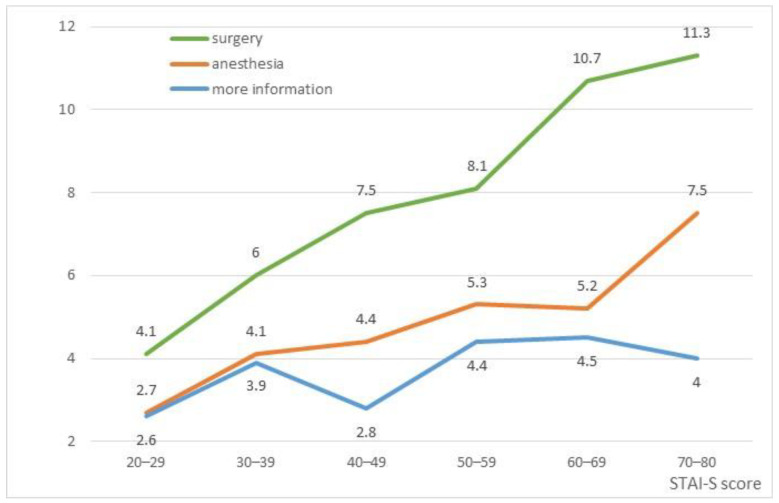
Trends of APAIS components along STAI-S scores.

**Figure 3 healthcare-11-02039-f003:**
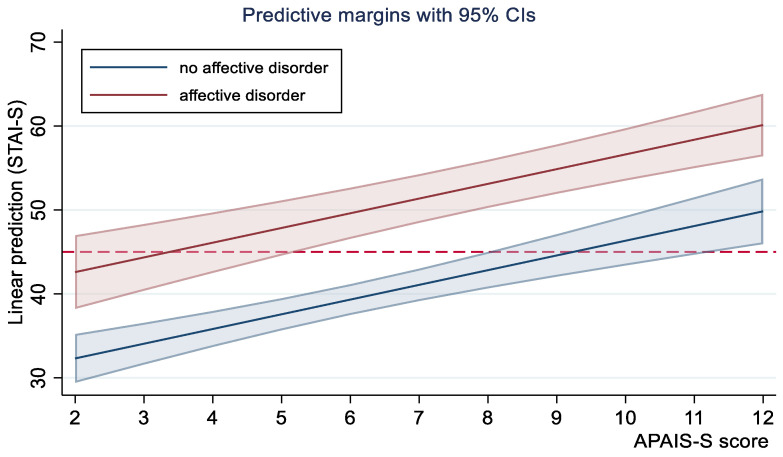
Mediating effect of APAIS-S (surgery procedure component) on preoperative anxiety (STAI-S) by risk groups. The horizontal dotted line represents the cut-off indicated by Bunevicious (≥45).

**Table 1 healthcare-11-02039-t001:** Preoperative state anxiety (STAI-S) by socio-demographic and clinic characteristics.

	N	Mean	sd	Min	Max	t/F (*p*-Value)
Total sample	104	43.3	12.9	20	77	
Gender						1.58
Male	47	41.1	13.7	20	74	(0.12)
Female	57	45.1	12.1	24	77	
Age						0.84
<50	11	46.0	12.0	30	68	(0.43)
51–69	64	42.0	12.7	20	74	
>70	29	45.1	13.8	24	77	
Education						1.04
Primary/middle	47	44.7	14.5	20	77	(0.30)
high/degree	57	42.1	11.5	22	70	
Marital status						0.28
Married/with partner	18	44.1	13.6	24	77	(0.78)
Single/Separated/widowed	86	43.1	12.9	20	74	
Working condition						0.44
Employed/student	37	42.5	11.6	22	70	(0.66)
Other	67	43.7	13.6	20	77	
Region						0.01
Veneto	18	43.2	15.9	24	74	(0.99)
Other regions	86	43.3	12.4	20	77	
Clinical pathway						2.5
Malignant upfront	51	42.5	13.5	22	74	(0.09)
Malignant chemio	27	40.4	10.6	20	62	
Benign + uncertain behaviour	26	47.8	12.3	28	77	
Type of intervention						1.06
DCP–whipple	53	43.8	14.3	20	77	(0.37)
exploratory laparotomy	13	38.3	12.1	22	61	
partial S-P resection	26	45.7	10.6	32	70	
total S-P resection	12	41.3	11.6	29	65	
STAI-T						12.59
Absent (20–39)	87	41.1	11.6	20	74	(<0.01)
Mild (40–49)	10	51.1	12.9	35	70	
Moderate + Serious (≥50)	6	63.3	11.8	45	77	
PHQ9						8.00
Absent (<4)	59	39.7	11.3	20	70	(<0.01)
Mild (5–9)	34	46.5	13.1	24	74	
Major depression (>10)	10	54.4	14.1	37	77	

S-P = spleno-pancreatic.

**Table 2 healthcare-11-02039-t002:** Preoperative worries/concerns (APAIS subscales) by socio-demographic and clinical characteristics.

	APAIS-Anaesthesia			APAIS-Surgery			APAIS-Information		
	M	sd	t/F (*p*)	M	sd	t/F (*p*)	M	sd	t/F (*p*)
Total sample	4.4	3.1		7.1	3.3		3.5	2.1	
Gender			1.83			1.73			1.38
Male	3.7	3.2	(0.07)	6.5	3.5	(0.09)	3.2	1.6	(0.17)
Female	4.9	3.0		7.6	3.1		3.8	2.4	
Age			0.05			0.35			0.27
<50	4.5	3.2	(0.95)	7.9	3.4	(0.70)	3.5	1.7	(0.77)
51–69	4.3	3.1		7.1	3.2		3.4	2.2	
>70	4.4	3.2		6.9	3.6		3.8	2.0	
Education			0.67			0.73			0.91
Primary/middle	4.1	2.9	(0.50)	7.4	3.4	(0.48)	3.7	2.1	(0.36)
high/degree	4.5	3.3		6.9	3.3		3.4	2.0	
Marital status			0.13			0.66			0.08
Married/with partner	4.4	2.7	(0.90)	7.6	3.0	(0.51)	3.6	2.4	(0.93)
Single/Separated/widowed	4.3	3.2		7.0	3.4		3.5	2.0	
Working condition			0.12			0.91			1.92
Employed/student	4.4	3.1	(0.98)	7.5	3.2	(0.37)	3.0	1.5	(0.06)
Other	4.3	3.2		6.9	3.3		3.8	2.3	
Region			0.53			0.03			0.21
Veneto	4.0	2.7	(0.60)	7.2	0.9	(0.98)	3.6	2.1	(0.84)
Other regions	4.4	3.2		7.1	3.3		3.5	2.1	
Clinical pathway			2.21			1.21			2.31
Malignant upfront	4.4	3.3	(0.11)	6.9	3.3	(0.30)	3.4	2.0	(0.10)
Malignant chemio	3.4	1.9		6.7	3.3		3.1	1.3	
Benign + uncertain behaviour	5.2	3.6		8.0	3.4		4.2	2.7	
Type of intervention			1.95			1.08			1.56
DCP–whipple	4.1	3.0	(0.13)	7.5	3.2	(0.36)	3.3	1.9	(0.20)
exploratory laparotomy	3.0	1.6		5.7	3.8		3.2	1.8	
partial S-P resection	5.4	3.7		7.3	3.3		4.3	2.6	
total S-P resection	4.7	3.0		6.8	3.1		3.3	1.6	
STAI-T			4.28			4.43			7.31
Absent (20–39)	4.0	2.8	(0.02)	6.7	3.2	(0.01)	3.3	1.7	(<0.01)
Mild (40–49)	5.7	4.5		9.0	3.9		3.4	2.1	
Moderate + Serious (≥50)	7.2	3.4		9.8	2.1		6.3	3.9	
PHQ9			3.23			2.61			1.9
Absent (<4)	3.7	2.8	(0.04)	6.5	3.2	(0.08)	3.2	1.7	(0.15)
Mild (5–9)	4.9	3.2		7.9	3.2		3.7	2.4	
Major depression (>10)	6.1	4.2		8.3	3.9		3.5	2.0	

M = mean; sd = standard deviation; t/F (*p*) = Student’s *t*-test or Fisher test (statistical probability); S-P = spleno-pancreatic.

**Table 3 healthcare-11-02039-t003:** Path model reporting the significant relationships for Preoperative state anxiety (STAI-S).

Dependent Variable	Independent Variable	Unstd Coef (Std Err)	*z* Test	*p*-Value	95% CI	Std Coeff
STAI-S	APAIS-S	1.75 (0.27)	6.38	<0.01	1.21–2.29	0.45
	STAI-T	0.45 (0.09)	4.84	<0.01	0.27–0.64	0.31
	PHQ9	0.54 (0.19)	2.82	<0.01	0.16–0.91	0.17
	const	14.32 (2.99)	4.78	<0.01	8.44–20.17	1.11
APAIS-S	STAI-T	0.14 (0.03)	4.52	<0.01	0.08–0.20	0.37
	const	2.72 (1.03)	2.63	<0.01	0.70–4.74	0.82
APAIS-I	STAI-T	0.07 (0.03)	2.83	<0.01	0.02–0.13	0.33
	const	1.09 (0.80)	1.37	0.17	−0.47–2.65	0.54
APAIS-A	PHQ9	0.23 (0.08)	2.96	<0.01	0.08–0.40	0.31
	const	3.27 (0.40)	8.08	<0.01	2.48–4.07	0.05
mean	STAI-T	31.39 (0.87)	35.86	<0.01	29.68–33.11	3.57
mean	PHQ9	4.47 (0.40)	11.07	<0.01	3.68–5.26	1.10
cov	STAI-T; PHQ9	14.15 (3.69)	3.83	<0.01	6.92–21.38	0.40
var	STAI-S	81.34 (11.69)			61.37–107.81	0.41
var	APAIS-S	9.47 (1.04)			7.62–11.76	0.86
var	APAIS-I	3.59 (0.63)			2.55–5.06	0.90
var	APAIS-A	8.77 (1.21)			6.69–11.49	0.90
var	STAI-T	77.40 (14.40)			53.75–11.45	1
var	PHQ9	16.46 (2.62)			12.06–22.48	1

## Data Availability

The data that support the findings of this study are available on request from the corresponding author. The data are not publicly available due to privacy or ethical restrictions.

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
