# Peer review of "Preoperative Anxiety in Patients with Pancreatic Cancer: What Contributes to Anxiety Levels in Patients Waiting for Surgical Intervention"

_healthcare, 2023, doi:10.3390/healthcare11142039_

Round 1

Reviewer 1 Report

- The article extensive English language proofreading by experts in the medical field.

- Study design should be mentioned in the abstract.

- In the method section, it was mentioned that "Three investigators (VM, DB and RS) selected those RCT participants who, having 89 similar diagnosis and clinical conditions, were expected to receive the same therapeutic 90 treatment and routine preoperative management the day before surgery, which included 91 a visit to the surgeon who delivered information and to the clinical psychologist who pro-92 vided support and indications on how to manage anxiety and main concerns." which increase the possibility of selection bias.

- Sample size calculation should be highlighted as the sample size is small.

- The authors should test the normality of the score before using the mean (sd). If it was not normally distributed, then they should use the nonparametric tests.

- I cannot see the regression analysis findings.

Author Response

Comments and Suggestions for Authors

- The article extensive English language proofreading by experts in the medical field.

--> The article has been revised by a mother tongue expert in the medical field (prof. Jerry Humphries –see acknowledgments)

- Study design should be mentioned in the abstract.

-->we added in the abstract that the study was “observational” (line 13)

- In the method section, it was mentioned that "Three investigators (VM, DB and RS) selected those RCT participants who, having similar diagnosis and clinical conditions, were expected to receive the same therapeutic  treatment and routine preoperative management the day before surgery, which included a visit to the surgeon who delivered information and to the clinical psychologist who provided support and indications on how to manage anxiety and main concerns." which increase the possibility of selection bias.

--> we thank the reviewer for this observation. To show that there was no bias selection due to researchers’ activity, we eliminated this misleading sentence and added a flow-chart in supplementary material (Figure 1S) describing the whole population attending the Pancreas Institute of the University Hospital of Verona between June 2017 and June 2018 (target period of the study) on the basis of intervention type (source: administrative data provided by the Pancreas Institute). The first part of this flow chart was already reported in Del Piccolo et al. 2021. Here we added the final selection of the study sample included in this paper.

- Sample size calculation should be highlighted as the sample size is small.

--> Thank you for raising this relevant and complex issue on SEM power analysis. We largely rewritten the statistical analysis section, to evidence the part of preliminary bivariate explorations and the later multivariate explorations, based on SEM approach. We added a sentence explaining the power test for model comparison, based on path analysis (lines 174-176): “Given the small dimension of our sample, we also performed a power calculation us-ing Power4SEM, a free R package [36] (see figure 3S in Supplementary material).”

 - The authors should test the normality of the score before using the mean (sd). If it was not normally distributed, then they should use the nonparametric tests.

-> We thank the reviewer for this suggestion. We performed the non-parametric median test for equal distribution for each variable, that confirmed the results included in tables 1 and 2 (see attachment-Median test for Rev 1). We also added skewness and Kurtosis values in tables 1S and 2S (supplementary material),

- I cannot see the regression analysis findings.

-->To satisfy this request, we decided to substitute figure 3 with a new table (Table 3).

Reviewer 2 Report

In this study, the authors describe in patients undergoing major surgery for (at least probable) pancreatic malignancy what influence trait anxiety and depression 1 month prior to surgery have on state anxiety on the day before surgery and with the concrete anxieties and fears with which this state anxiety is in turn associated. This referent does not consider himself qualified to evaluate the statistical methods used for value and accuracy. The significance of this study is low: the numbers are relatively small and the findings unspectacular. It is recommended that the article be greatly shortened to a short communication and that most of the tables and figures be moved to the supplementary material. Reviewer also objects to the last sentences of the Abstract "Findings can [...] preoperative anxiety." and the last paragraph of Conclusions "Findings from [...] communications and practices." These state subjects that have not been studied at all and are far from certain.

Minor comments are:

- The abbreviations STAI-S and APAIS should also be written out for the first time in the abstract.

- Please provide the confidence intervals of the listed percentages in the Introduction.

- Please provide clarity about the pool of patients from which participants are selected and reasons why some of it were excluded. This should also be discussed in the discussion.

- To what extent did the patient population of Bunevicius et al. (line 112) have the same (age and gender) composition as the current one (line 170)?

- That the modification of the APIAS does not affect the characteristics of this scale (lines 132 to 134) should be described further and in the discussion.

- What is the significance of 15 (14%) of patients being outside the tolerance limit (line 175)?

- In the text, please check the use of the correct decimal point (period instead of comma). This is especially true for Figure 3.

- The rationale for not correcting for gender belongs in the discussion and is too limited and therefore inadequate.

- The discussion should be rewritten in the opinion of this reviewer. First, the design and results should be stated, then the methods used should be thoroughly discussed, then the connection or lack of connection to the literature should be described, then the context in which the results should be placed, and finally the advantages and limitations. Reviewer appreciates the attempt to improve the care of their own patients through this study, but the population size is small and the findings limited. Detailed elaboration on the significance and implications is therefore not appropriate in this publication.

Author Response

Comments and Suggestions for Authors

In this study, the authors describe in patients undergoing major surgery for (at least probable) pancreatic malignancy what influence trait anxiety and depression 1 month prior to surgery have on state anxiety on the day before surgery and with the concrete anxieties and fears with which this state anxiety is in turn associated. This referent does not consider himself qualified to evaluate the statistical methods used for value and accuracy.

The significance of this study is low: the numbers are relatively small and the findings unspectacular.

--> We thank the reviewer for the considerations regarding the robustness of our results and their possible impact, since this is a really important issue. We agree that our sample is relatively small. This is due to a set of reasons strictly related to our object of interest. The first is that pancreatic malignancies are rarer than other malignancies. Beside this, only a small percentage (20%) of the new diagnosis, is eligible for surgery. Finally, considering the peculiar psychophysical condition of these patients even a smaller part of them remain eligible until the day before surgery and/or available for a psychological assessment. These elements justify the paucity of studies that explored psychological functioning of this population and outlines the relevance of our results. We have better highlighted these elements in the discussion (lines 382-388): “Patients undergoing emergency surgery were not included. Only elective surgical participants and only those who could be evaluated by the psychologist the day before surgery were considered. But we have also to remember that upfront surgical resection of pancreatic cancer is feasible in less than 20% of patients at diagnosis [47], therefore high numbers in this kind of sample are difficult to reach. For this reason, future studies need to include different hospital settings”.

Regarding relevance, we think that another strength of our study is that it provides practical indications on variables involved in the development of preoperative anxiety not only for psychologists but also for surgeons involved in the healthcare process (lines 353-363): “This suggests that an appropriate training of surgeons in communications skills and a correct identification of the main topics to discuss during preoperative phases may contribute to possibly lower higher levels of global state anxiety. This seems to be fur-ther confirmed by the results emerging from multivariate regression and path analysis, which showed that trait anxiety (STAI-T) and depressive symptoms (PHQ-9) influ-enced preoperative concerns (APAIS) and accordingly, preoperative state anxiety (STAI-S). It has been suggested that reducing anxiety during a consultation may lead to better retention of information, a stronger physician-patient relationship and, ulti-mately, to enhance well-being [43]. Physicians need to be able to use effective commu-nication skills: an effective use of assessment, information and supportive skills can contribute to reduce patient anxiety [44].”

It is recommended that the article be greatly shortened to a short communication and that most of the tables and figures be moved to the supplementary material. Reviewer also objects to the last sentences of the Abstract "Findings can [...] preoperative anxiety." and the last paragraph of Conclusions "Findings from [...] communications and practices." These state subjects that have not been studied at all and are far from certain.

We moved from main text and added several tables and figures in supplementary material section, but given the complexity of the study procedure and the statistics adopted, we think that we are not able to shorten more the paper.

Minor comments are:

- The abbreviations STAI-S and APAIS should also be written out for the first time in the abstract.

--> we changed the text accordingly to the suggestion

- Please provide the confidence intervals of the listed percentages in the Introduction.

--> We are sorry but, only one out of the 4 cited papers, provides the confidence intervals. As requested we added this value in the text (see lines 43-44)

- Please provide clarity about the pool of patients from which participants are selected and reasons why some of it were excluded. This should also be discussed in the discussion.

--> we thank the reviewer for this observation. Considering the high attrition rate that characterizes this population, we have better explained in the text the selection process that lead to the present sample size by adding a flow-chart in supplementary material (Figure 1S) that describes the whole population attending the Pancreas Institute of the University Hospital of Verona between June 2017 and June 2018 (target period of the study) on the basis of intervention type (source: administrative data provided by the Pancreas Institute). The first part of this flow chart was already reported in Del Piccolo et al. 2021. Here we added the final selection of the study sample included in this paper.

We also added in the discussion this limitation (lines 383-388): “Only elective surgical participants and only those who could be evaluated by the psychologist the day before surgery were considered. But we have also to remember that upfront surgical resection of pancreatic cancer is feasible in less than 20% of patients at diagnosis [47], therefore high numbers in this kind of sample are difficult to reach. For this reason, future studies need to include different hospital settings.”  and provided the references of our previous publications in the Methods section.

- To what extent did the patient population of Bunevicius et al. (line 112) have the same (age and gender) composition as the current one (line 170)?

--> thank you for rising this point. Indeed, we chose this sample as a reference because it was a clinical sample (cardiac patients), more comparable to our sample than general population, even if some differences were present: characteristics of the sample described in Bunevicius et al:  Mean age 57.5, s.d. 9.2 median 57; gender 78% male. Our sample had a mean age of 63, s.d. 11.5 median 65 and 45% were males.

- That the modification of the APAIS does not affect the characteristics of this scale (lines 132 to 134) should be described further and in the discussion.

--> We specified in the methods section (lines 136-138) that to check the internal consistency of the scales we performed Cronbach's Alpha (results were already reported in the manuscript in table 1A-Appendix A, now they are reported in Table 2S in Supplementary material): “To check the internal consistency of the scales we performed Cronbach's Alpha (see Table 2S in Supplementary material).” and discussed results in lines 283-287 of the discussion: “This choice has not led to significant consequences in terms of internal consistency (as evidenced by Cronbach's Alpha indexes which were generally acceptable or good). To-tal Cronbach's Alpha in our study was fairly good (Cronbach's Alpha 0,73) even if slightly lower that that reported by the author of the Italian validation [29] (Cronbach's Alpha 0,84) on a sample of 110 patients undergoing elective surgery.”

- What is the significance of 15 (14%) of patients being outside the tolerance limit (line 175)?

--> Ilardi identifies the 90th percentile of Italian elderly population indicating a cut-off score of 61 (Ilardi et al. 2021). In our sample 15 patients (14%) presented STAI-S scores equal or higher than this score, suggesting severe anxiety. We specified this at lines 115-118: “Moreover, due to high mean age of our sample, we also adopted Ilardi et al [25] cut-off score>61 on elderly population as indication for a score outside the tolerance limit (90th percentile – severe anxiety).”

- In the text, please check the use of the correct decimal point (period instead of comma). This is especially true for Figure 3.

--> Thank you for rising this point. We checked the text and corrected accordingly.

- The rationale for not correcting for gender belongs in the discussion and is too limited and therefore inadequate.

--> Our primary outcome is preoperative anxiety (measured through STAI-S and APAIS). Preliminary results indicate no significant differences in STAI-S and APAIS subscales scores related to gender, therefore we reported the following (lines 229-234): “As indicated in Tables 1 and 2, gender, age and education showed no significant impact on STAI-S and APAIS scores. A difference related to gender was only observed on trait anxiety (STAI-T), with higher mean scores in the female group (mean value 33.2, sd 9.3, vs male: 29.1, sd. 7.7, t=2.45 p=0.02), therefore we preferred not to account for gender confounding effect in path analysis models, leaving gender differences subsumed under trait anxiety (STAI-T) scores.”

In the discussion we than reported the same concept, adding some considerations: “Regarding gender, age and education, no significant differences were found in our primary outcomes related to preoperative anxiety (STAI-S and APAIS scores). A difference related to gender was observed only on trait anxiety (STAI-T), with higher mean scores in the female group. Nevertheless, gender did not affect directly STAI-S and APAIS scores, therefore we preferred not to account for gender confounding effect in path analysis models, given that gender differences could be subsumed under trait anxiety (STAI-T) scores. This has also practical implications: thinking to a screening activity during day-hospital stay, it is preferable to stick to the use of specific screening tests rather than to predict outcomes on the basis of personal characteristics such as gender, with the risk of suggesting evaluations based on gender stereotypes.” (lines 318-327)  

- The discussion should be rewritten in the opinion of this reviewer. First, the design and results should be stated, then the methods used should be thoroughly discussed, then the connection or lack of connection to the literature should be described, then the context in which the results should be placed, and finally the advantages and limitations. Reviewer appreciates the attempt to improve the care of their own patients through this study, but the population size is small and the findings limited. Detailed elaboration on the significance and implications is therefore not appropriate in this publication.

--> We thank the reviewer for the appreciation of our efforts. We rewrote most of the discussion according to reviewer’s suggestions and hope that now it corresponds to what expected.

Reviewer 3 Report

Thank you for inviting me to review this manuscript. Overall it is interesting. It seems to me that this is a paper submitted to Healthcare but in the manuscript it has logos for IJERPH. 

The theoretical contribution seems to be unclear to me. It seems very obvious that anxiety traits contribute to anxiety. It would be helpful for authors to elaborate the theoretical reasoning for this investigation in the introduction. 

Lines 321-323 seems to be too short for a paragraph, considering combine it. Please review the whole paper to avoid using single sentence as a paragraph. 

Line 57, it should be amount of information rather than quantity of information. 

It is important to acknowledge that other psychological characteristics such as personality traits are not assessed in this study, as personality traits is related to mental health (e.g., Kang, W., Steffens, F., Pineda, S., Widuch, K., & Malvaso, A. (2023). Personality traits and dimensions of mental health. Scientific Reports13(1), 7091.)

Line 35 please remove "as know." Please remove this phrase throughout the whole manuscript as this is incorrect. 

Author Response

Comments and Suggestions for Authors

Thank you for inviting me to review this manuscript. Overall it is interesting. It seems to me that this is a paper submitted to Healthcare but in the manuscript it has logos for IJERPH. 

--> Yes, we originally submitted the manuscript to IJERPH that, following an agreement between the two journals, automatically transferred our manuscript to Healthcare.

The theoretical contribution seems to be unclear to me. It seems very obvious that anxiety traits contribute to anxiety. It would be helpful for authors to elaborate the theoretical reasoning for this investigation in the introduction. 

--> we really thank the reviewer for this suggestion that allowed us to better elaborate and clarify in the text our line of reasoning.

Even if this is one of the few papers where emotional needs of patients with pancreatic malignancies are explored in depth, we agree that it is quite obvious that trait anxiety contributes to state anxiety. What we consider more meaningful about our hypothesis (confirmed also by our results), is that we explored whether trait anxiety can lead to different preoperative concerns, and whether these concerns may have a mediating role on preoperative state anxiety. This aspect has implications for all the healthcare providers involved in the caring process: psychologist can early screen patients at higher risk of anxiety and surgeons/ anesthesiologists can tailor the communication process (in terms of time and content) on patients’ preoperative concerns (fostering an integrated approach to health care and optimizing the available resources).

In order to better clarify this line of reasoning we have:

  1. Changed Figure 1 in the Introduction, trying to be more explicit on the line of reasoning we followed.
  2. Introduced two possible models of path analysis, to better understand the relationship between STAI-T, PHQ-9, APAIS components and STAI-S scores (lines 162-168): “Two models were then elaborated: 1. a basic model (H0 hypothesis) which fitted just the direct effect of preoperative concerns (APAIS) on preoperative state anxiety (STAI-S), considering STAI-T and PHQ, as covariates; 2. a second model (H1 hypothe-sis) which assumes that preoperative concerns (APAIS) mediate the effects of patient psychological characteristics (STAI-T and PHQ) on preoperative state anxiety (STAI-S) (see figure 2S in supplementary material).”

Lines 321-323 seems to be too short for a paragraph, considering combine it. Please review the whole paper to avoid using single sentence as a paragraph. 

--> we have corrected the text accordingly.

Line 57, it should be amount of information rather than quantity of information. 

--> we have corrected the text accordingly.

It is important to acknowledge that other psychological characteristics such as personality traits are not assessed in this study, as personality traits is related to mental health (e.g., Kang, W., Steffens, F., Pineda, S., Widuch, K., & Malvaso, A. (2023). Personality traits and dimensions of mental health. Scientific Reports13(1), 7091.)

--> Thank you for rising this point. We added this aspect among our study limitations (lines 389-393): “Another limitation relates to the variables included as predictors. Other psycho-logical characteristics, such as personality traits were not assessed in this study. Be-neath they are related to mental health [48], we have to consider that the study was based on screening activities conducted during day-hospital stay, therefore the analysis of personality characteristics was beyond the scope of a routine hospital activity.”

Line 35 please remove "as know." Please remove this phrase throughout the whole manuscript as this is incorrect. 

--> we have corrected the text accordingly.

Round 2

Reviewer 1 Report

No further comments.

Reviewer 2 Report

This reviewer regrets that the authors did not follow his advice to shorten the manuscript to a short communication.